# A Cross-Sectional Survey of Pediatric Infectious Disease Physicians’ Approach to Congenital Cytomegalovirus Infection [note 1]

**DOI:** 10.3390/ijns9020017

**Published:** 2023-03-24

**Authors:** Chieko Hoki, Michelle White, Megan H. Pesch, Ann J. Melvin, Albert H. Park

**Affiliations:** 1Department of Otolaryngology- Head and Neck Surgery, University of Utah School of Medicine, Salt Lake City, UT 84132, USA; 2Division of Developmental and Behavioral Pediatrics, Department of Pediatrics, University of Michigan Medical School, Ann Arbor, MI 48109, USA; 3Pediatric Infectious Disease, Seattle Children’s Hospital, University of Washington, Seattle, WA 98105, USA

**Keywords:** CMV, congenital, cytomegalovirus, infection, screening, newborn, infectious disease, torch infection

## Abstract

Congenital cytomegalovirus (cCMV) continues to be a major public health care issue due to its high prevalence throughout the world. However, there is a paucity of studies evaluating how providers manage this infection. This study surveyed North American Pediatric Infectious Disease (PID) physicians to elicit their approach towards the evaluation and treatment of this condition. Thirty-two PID physicians responded to this survey. Institutional testing and screening for cCMV were infrequently reported. The respondents in general agreed upon most laboratory and diagnostic testing except for neuroimaging. For those tests, there was a disparity in indications for head ultrasound versus brain MRI imaging. Most (68.8%) agreed with the clinical practice of starting valganciclovir in an infant less than 1 month of age with one sign or symptom of disease, and 62.5% would do so for an infant with isolated sensorineural hearing loss. However, only 28.1% would treat cCMV-infected infants older than 1 month of age. In conclusion, few healthcare institutions represented by PID physicians in this cohort had a cCMV screening or testing initiative, yet most respondents would test at a much higher level based on their clinical practice. While there is general consensus in evaluation and treatment of these children, there are disparities in practices regarding neuroimaging and indications for antiviral treatment with respect to age and severity of disease. There is a great need for an evidence based policy statement to standardize cCMV workup and treatment.

## 1. Introduction

Congenital cytomegalovirus (cCMV) is the most common infectious birth defect worldwide [1] It has an estimated incidence rate of 0.6–0.7% of all live births in the US [2,3] The signs and symptoms of cCMV can range from jaundice, petechiae, hepatosplenomegaly, and central nervous system involvement (e.g., microcephaly, intracranial calcifications, seizures) in the more severe or symptomatic cases, to no signs at all in asymptomatic cases [4] The most common sequelae from this condition is sensorineural hearing loss (SNHL), which develops in 33% of symptomatic and 10% of asymptomatic infants [5] In 2017, two sets of expert recommendations and consensus guidelines for the diagnosis and treatment of cCMV were published, including one from the International Cytomegalovirus Consensus group, which included top North American infectious disease clinicians in the field [6,7] While many guidelines align in terms of recommended treatment, controversy remains about which infants should be treated with antiviral medication (i.e., valganciclovir or ganciclovir). One reason for this inconsistency is the relatively few clinical trials evaluating antiviral treatment efficacy and safety among infants across the spectrum of cCMV severity [8,9]

To date, there has been little research examining the clinical practice patterns of North American healthcare providers caring for infants with cCMV. While prior studies have reported the cCMV fund of knowledge amongst speech and language pathologists [10], otolaryngologists [11], pediatricians [12], and audiologists [13], no studies to date have examined the clinical practices of pediatric infectious disease (PID) physicians. Understanding the trends in clinical practice among those clinicians most likely to treat cCMV with antiviral medication and oversee the diagnostic workup is critical to gaining insights about best practices and how to optimally treat these patients. Therefore, the objective of this study was to examine cCMV knowledge and clinical practices from healthcare institutions across North America.

## 2. Methods

Participants and recruitment. A survey was sent initially to the PID society newsletter (n = 195 physicians) and then to PID physicians involved in the NIH-funded ValEAR clinical trial (ValEAR trial ClinicalTrials.gov NCT03107871). The study included 29 PID physicians from 34 institutions across the US (n = 30) and Canada (n = 4). Additional PID physicians were contacted at a CMV public policy health and policy conference held in Ottawa.

Survey design. The surveys were created through Google Forms (see Appendix A), which included a brief description and purpose of the study. There were three parts to the survey: (a) demographics, (b) background knowledge of cCMV, and (c) cCMV testing and treatment. The demographics section included provider affiliation with any academic institution, age, gender, years in practice, race/ethnicity, training specialty, and percent of patients that are insured through Medicaid in their practice. The background knowledge of cCMV section identified current knowledge of signs and symptoms of disease, effect on hearing loss, transmission, and diagnostic methods. The cCMV testing and treatment section examined the provider’s approach to disease testing and screening, and current practice of treatment. The responses were collected between January and September of 2022. This study was deemed exempt from Institutional Review Board approval (IRB # 00146185).

## 3. Results

### 3.1. Demographics

A total of 32 PID physicians practicing or training at 14 different hospitals or institutions in the US and Canada completed the survey (Table 1). The number of PIDs represented by each institution ranged from 1 to 6. Eleven PIDs did not indicate their institutional affiliation. The average age of participants was 49 ± 10 years of age (range 30–66 years), with 53% of the respondents identifying as male. Demographically, all of the respondents identified as either Asian (21.9%) or White (81.3%). Half of the respondents had been in practice for more than 15 years (50%). Over, 90% of respondents identified as being in academic practice or training in an academic institution; most (n = 11; 61.1%) participants practiced at an institution where <50% of the patients were covered by Medicaid insurance.

### 3.2. Knowledge of cCMV Symptoms and Diagnosis

A majority of respondents identified the signs and symptoms of cCMV, with all 32 (100%) respondents correctly noting its association with hearing loss and petechiae, 30 (93.8%) with intellectual disability and loss of vision and 28 (87.5%) with seizures. All respondents correctly identified that oral cavity ulcers are not associated with cCMV (Table 2).

Knowledge of CMV and hearing was almost as high. Over eighty percent were aware that approximately 10% of children with asymptomatic cCMV develop SNHL, but less than 50% (37.5%) knew that 33% develop SNHL with symptomatic infection (Table 3). Only a few (15.6%) knew that of children with cCMV associated SNHL, 20% develop progressively worsening SNHL based on severity of infection (Table 4).

Almost all respondents (96.9%) recognized that urine PCR or culture prior to 3 weeks of age is an effective method of diagnosing cCMV (Table 5). Fewer agreed about the benefits from neonatal dried blood spot polymerase chain reaction (DBS CMV PCR) testing for an older child with possible CMV (12.5%) or selected a response concerning the potential false positive outcomes from saliva CMV PCR testing (21.9%). All recognized that serologic testing of IgG is not an effective method. When asked specifically which test(s) can definitively establish a diagnosis for cCMV in children greater than 3 weeks of age, 68.8 % correctly identified that dried blood spot CMV PCR was the correct method of diagnosis (Table 6). Approximately twenty-two percent of respondents incorrectly chose imaging studies such as CT or MRI as a definitive modality for diagnosis. One mentioned eye findings and another stated that a history of maternal seroconversion during pregnancy in an infant with signs or symptoms consistent with cCMV. The vast majority (96.9%) noted that if a child with SNHL had an “undetectable” CMV reading on dried blood spot, he or she may still have a cCMV infection.

Knowledge of the methods of transmission of CMV was in general strong with nearly all recognizing that kissing, changing diapers without washing hands afterwards, drinking breast milk, and receiving a blood transfusion are methods of horizontal transmission (Table 7). Seventy-eight percent recognized that sexual intercourse is a method of transmission. The least recognized modality was sharing food, with 62.5% respondents correctly identifying it as a mode of transmission.

### 3.3. Screening and Testing at the Institutional and Physician Level

Overall, 34.4% of institutions had no protocol for neonatal cCMV screening or testing, though several had targeted screening programs (Figure 1). Of the targeted cCMV screening programs, signs that most often triggered screening were for failed newborn hearing screening (n = 18, 56%), SNHL (n = 15, 47%), microcephaly (n = 13, 40%) or idiopathic thrombocytopenia (n = 12, 37.5%). Other indications included petechial rash (n = 10, 31.3%), hepatosplenomegaly (n = 9, 28.1%), being small for gestational age (n = 10, 31.3%), or a history of maternal CMV infection (n = 8, 25%).

Respondents tended to screen or test more neonates for cCMV than their institutional protocol. Only 9.4% did not test or screen for cCMV in their clinical practice (Figure 1). Of respondents who did advocate targeted screening/testing for cCMV, the most common indications prompting testing were SNHL (n = 25, 78.1%) and microcephaly (n = 25, 78.1%). Other indications included idiopathic thrombocytopenia (n = 24, 75%), being small for gestational age (n = 17, 53.1%), failed newborn hearing screening (19, 59.4%), history of maternal CMV infection (n = 19, 59.4%), petechial rash (n = 24, 75%), and hepatosplenomegaly (n = 24, 75%). One respondent practices universal cCMV testing. Two have implemented universal cCMV screening for any infant in the neonatal intensive care unit.

### 3.4. Evaluation and Treatment of Patients with Confirmed cCMV Infection

Upon confirmation of a cCMV infection, nearly all respondents supported the practice of ordering a complete blood count with differential (n = 31/32, 96.9%), referring for an ophthalmology evaluation (n = 30/32, 93.8%), as well as diagnostic hearing testing (n = 32/32, 100%; Figure 2). Most (n = 24/32, 75%) recommended comprehensive metabolic profile, 59.4% (n = 19/32) would propose developmental services, and 50% (n = 16/32) would suggest early intervention services. Almost all (n = 21/22, 95.2%) respondents recommended ordering neuroimaging upon diagnosis. Of those that would order neuroimaging, 34.4% (n = 11/32) would order cranial ultrasound (HUS) for all cases, and 21.9% (n = 7/32) would order brain magnetic resonance imaging (MRI) for all cases (Table 8). A majority responded that they would order HUS followed by brain MRI if an abnormality was found, and 21.9% (n = 7/32) stated that they would order a HUS unless the infant presented with microcephaly, in which case they would order a brain MRI.

In general, most respondents would administer VGCV for a younger infant (less than one month versus older than 6 months of age) and with more severe disease (e.g., one or more signs or symptoms of cCMV infection). Over sixty-two percent would treat a cCMV-infected infant less than 1 month of age with isolated SNHL. Thirty-one percent would treat this child even if he or she were between 1 to 6 months of age. Less than 5% would treat a newborn with asymptomatic cCMV infection. Fifty-six percent would obtain an absolute neutrophil count weekly for 6 weeks, then every other week for a month, and then every month until treatment completed (Figure 3). Over eighty-seven percent would order a complete blood count (CBC) with differential or comprehensive metabolic profile (56.2%) during treatment (Table 9). Over ninety percent would administer VGCV for 6 months.

When asked what they would do in the case of an infected infant having an absolute neutrophil count of 300 cells/μL on treatment with VGCV, most responded that they would stop the drug for a week and test again, and 42.9% said that they would restart at the original dose. When asked what they would consider a treatment for cCMV, all considered VGCV a treatment and over 60% also considered serial hearing testing, early intervention, and hearing aids or cochlear implants as treatments.

There was no consensus on hearing follow-up surveillance for infants with cCMV, but the most common recommendation (n = 15, 46.9%) was every 3 to 6 months for the first year, then every 6 months until 3 years, then annually until age six.

## 4. Discussion

We report for the first time to our knowledge the results of a survey studying cCMV knowledge and clinical practices amongst PID physicians in institutions across North America. In general, we found that PID physicians possess a high fund of knowledge regarding cCMV infection in comparison to other providers [11,13] and with some exceptions an overall consensus on many aspects of evaluation and treatment.

### 4.1. Knowledge of cCMV Symptoms and Diagnosis

The survey revealed a high level of knowledge amongst PID physicians on the signs and symptoms of cCMV, methods of horizontal CMV transmission, and the most effective methods of diagnosis of cCMV infants under 3 weeks of age. There was a lower level of knowledge when it came to an understanding concerning disease progression especially as related to hearing loss, on other effective methods of diagnosis, and especially on methods of diagnosis in infants greater than 3 weeks of age.

One of the most concerning findings from our survey was the discrepancies in knowledge amongst PID physicians on how to most accurately diagnose cCMV. While 90.6% correctly identified urine PCR/culture prior to 3 weeks of age as an accurate mode of diagnosis, only 46.9% of our surveyed physicians recognized saliva CMV culture with urine PCR confirmation as another accurate form of diagnosis in children under 3 weeks of age. Studies by Ross et al. and Puhakka et al. have reported false positive rates between 7.5–26.7% when using saliva CMV testing, presumably from breast milk contamination [14,15]. Additionally, only 12.5% knew that dried blood spot CMV PCR could be used as a correct method of diagnosis in infants and children older than 3 weeks of age, and 18.8% of respondents correctly noted the limitations of saliva CMV PCR testing.

Previous studies examining pediatric otolaryngologists, audiologists, and speech language pathologists and their recognition of cCMV diagnostic and treatment modalities reported knowledge gaps in each group [10,11,13]. Specifically, otolaryngologists (53.2%) and audiologists (57.3%) fared better than speech and language pathologists (23.8%) in knowledge of cCMV transmission. The PID physicians examined in this study showed significantly higher levels of knowledge on transmission, knowledge of cCMV testing, knowledge of timing for cCMV diagnosis and knowledge of its relationship to SNHL compared to the other groups.

### 4.2. Screening and Testing at the Institutional and Physician Level

The responses indicate a relatively low level of cCMV testing or screening. Thirty-four percent of institutions had no early CMV screening or testing despite several consensus statements and papers reporting multiple benefits from early detection [6,7,16,17,18]. One such consensus statement came from an European group of Pediatric Infectious Disease physicians, where they recommended that all newborns with confirmed SNHL be tested for cCMV within the first 21 days of life [7]. Haller et al. reviewed the literature as it related to hearing-targeted early CMV testing and used the Wilson and Jungner criteria to evaluate this method of testing [18]. Based on these criteria, they found substantial rationale and evidence to support a hearing-targeted approach to testing for congenital CMV. Recently, Suarez et al. reported that an expanded targeted early cCMV testing program can significantly improve detection rates compared to a more limited hearing-targeted early CMV program [19]. Testing was carried out for any newborn found to meet any of the following criteria: maternal history of CMV infection, idiopathic elevated liver enzymes or bilirubin, failed hearing screening, abnormal central nervous system imaging findings suggestive of cCMV, being small for gestational age, microcephaly, unexplained hepatosplenomegaly or petechial rash. Many of these criteria were cited for testing by the PIDs in this survey.

This low institutional testing may be why, when asked about their own personal practices regardless of their institutional protocols, many of our surveyed physicians advocated greater screening or testing. Seventy-eight percent of our surveyed physicians personally test for cCMV in newborns with diagnosed SNHL, which is significantly higher than the 47% of institutions without a cCMV testing protocol. This difference was also seen for maternal infection, idiopathic thrombocytopenia, petechial rash, and small for gestational age. The rationale for their higher rate of testing was not asked in this survey. Perhaps their clinical practice and referral patterns would result in more testing. Certainly, implementing an institutional protocol for early CMV testing is a significant undertaking that requires support from many providers and staff, a process to ensure testing and appropriate follow-up. This task would be aided by a position statement from the American Academy of Pediatrics.

### 4.3. Evaluation and Treatment of Patients with Confirmed cCMV Infection

Overall, PID physicians agreed on the need for CBC with differential, ophthalmology evaluation, and a diagnostic hearing test. Surprisingly, there was less consensus (only 50%) on the need for early intervention services. In addition, all agreed on the need for neuroimaging, but there was a disparity on whether to start with a cranial ultrasound or brain MRI. This result may not be surprising given the reported discordant findings between HUS and brain MRI [20], and the greater cost and potential need for sedation from MRI brain imaging [20,21]. That imaging should be done is supported by a study by Hranlovich et al. noting almost 60% of cCMV-infected infants identified by a hearing-targeted early CMV testing approach had abnormal brain MRI findings [22]. Capretti et al. found three cCMV-infected infants with normal head ultrasound but abnormal MRI scans out of 40 who underwent both procedures [21]. Two of these infants were later found to have psychomotor delay; the other had an attention deficit disorder. Thus, relying on just a head ultrasound could delay identification of cCMV-infected children at risk for psychomotor delay. Smiljkovic et al. also reported on discordant results between head ultrasound and MRI imaging [23]. Of concern was that two cCMV-infected infants with apparently abnormal head ultrasounds underwent VGCV treatment before a subsequent normal brain MRI was performed.

VGCV has been shown to provide improved hearing and neurocognitive outcomes when administered to symptomatic cCMV-infected infants less than 1 month of age [9,24]. Thus, it is not surprising that 68.8–78.1% of PID physicians would advocate for antiviral therapy for a child with severe disease. Ninety-percent of our surveyed physicians chose 6 months of treatment, none chose a duration less than 6 months, and 3.1% chose to treat for longer than 6 months. There are a paucity of studies evaluating longer than 6 month VGCV treatment for cCMV infections. Bilavsky et al. reported relatively favorable outcomes following different 12 month duration combinations of ganciclovir or VGCV in symptomatic cCMV infants with SNHL [25]. They found improved hearing outcomes in these infants and suggested that longer treatment than 6 months may be necessary, especially in children demonstrating severe SNHL; it should be noted, however, that their study did not include an untreated control group for comparison. It is interesting that 28.1–34.4% of respondents would initiate treatment if the child were between 1 and 6 months of age. Hopefully, the results from the VGCV Therapy in Infants and Children with cCMV Infection and Hearing Loss (NCT01649869) will provide needed information on the utility of this medication for older children.

The role of VGCV for cCMV-infected children with mild disease, asymptomatic or asymptomatic, with isolated SNHL is unclear. Over sixty percent of the respondents would administer VGCV for a cCMV-infected newborn with isolated SNHL. Pasternak et al. retrospectively studied 59 infants with cCMV and isolated SNHL treated within 4 weeks of life and continued for a year [26]. Improvement in hearing was noted in 69% of affected ears. The lack of a control group limits the interpretability of these results [27]. An NIH-funded, double-blind, placebo-controlled clinical trial (ValEAR) designed to determine the efficacy and safety of 6-month VGCV therapy in asymptomatic cCMV-infected infants 1–12 months of age with isolated SNHL (NCT03107871) was recently closed due to a lack of enrollment. A systematic review examining the existing evidence of VGCV on hearing in children with cCMV infection found insufficient evidence to support VGCV treatment of children with asymptomatic cCMV infection and isolated SNHL [28]. Future prospective randomized clinical trials to address this issue are needed.

Data addressing the role of VGCV in asymptomatic disease are scarce. Approximately 3% of the respondents would recommend VGCV for this group of patients. Lackner et al. evaluated the long-term hearing outcomes of 18 asymptomatic cCMV-infected children randomized to undergo GCV or no treatment [29]. Two of the eight untreated cCMV-infected children developed SNHL: one was detected at 8 years of age and the other at 10 years of age. All of the 10 children in the ganciclovir treated group had normal hearing. This small pilot study suggests that in asymptomatic cCMV-infected neonates, hearing deterioration in early childhood may be prevented by early (within the first week of life) intravenous administration of GCV or potentially by VGCV. Unfortunately, an NIH-funded phase II open-label trial to evaluate VGCV as a treatment to prevent development of SNHL in infants with asymptomatic cCMV infection was halted due a high rate of neutropenia in the treated subjects (personal communication, D. Kimberlin Feb 2022).

There is no uniform consensus on neutrophil and other laboratory surveillance while on VGCV therapy, or the timing or duration of hearing surveillance for these children [6,7]. The results of the survey reflect some disagreement on these issues. In general, however, most would recommend assessment of neutrophil counts and a comprehensive metabolic profile while on treatment. This recommendation would be supported by several studies reporting a high incidence of neutropenia and elevated transaminases in cCMV-infected infants on VGCV [8,9]. Hearing surveillance at least until the child is 6 years of age was recommended by the PID physicians. This result is consistent with Lanzieri et al.’s longitudinal study of asymptomatic cCMV-infected children [30]. They found that the risk of developing SNHL after age 5 years was no different than in uninfected children. These findings would not necessarily pertain to those with more severe cCMV infection and hearing loss. Goderis et al. noted that the proportion of those with bilateral severe to profound SNHL is higher in those with symptomatic than asymptomatic infection [5]. Even among those with less severe infection, Torrecillas et al. noted a significant proportion of cCMV-infected children with delayed onset SNHL continuing to experience progressively worsening hearing well into their adolescent years [31].

The biggest limitations of our study are our small sample size of 32 participants and the limited number of institutions. Only 14 institutions were represented in our survey, with the majority of these being from the US. Thus, these findings cannot be generalized and are likely not representative of PID practice in North America. Unlike other organizations (e.g., audiology, speech and language pathology or otolaryngology) which have straightforward means to contact their membership, surprisingly, the PID society does not seem to have a similar infrastructure. Since the respondents were de-identified, we could not be entirely certain that respondents produced duplicate answers. However, respondents from the same institution were evaluated and none provided identical answers suggesting that the respondents were different. Additionally, we are unable to determine whether the responses were selected based on actual knowledge of the question or mere guessing. Despite these limitations, we believe this study provides novel information on the scope of knowledge, areas of diagnostic and treatment consensus and discordance amongst PID physicians. There is a need to increase the number of respondents in a future survey.

In conclusion, very few healthcare institutions represented by PID physicians in this cohort have implemented any form of early CMV testing or screening, yet most respondents will test at a much higher level based on their clinical practice. While there is general consensus in evaluation and treatment of these children, there are disparities in practices regarding neuroimaging and indications for antiviral treatment with respect to age and severity of disease. A policy statement to address the role for early CMV testing and evaluation is needed to provide a more consistent evidence based optimal care for this vulnerable group.

## Figures and Tables

**Figure 1 IJNS-09-00017-f001:**
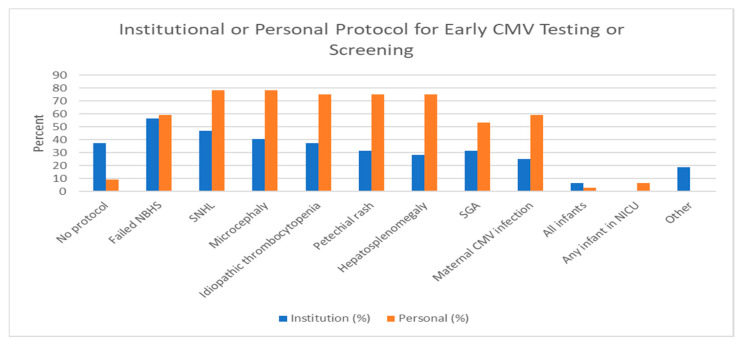
Institutional or Personal Protocol for Early CMV testing/screening. NICU: neonatal intensive care unit; SNHL: sensorineural hearing loss; SGA: Small for gestational age; NBHS: newborn hearing screen. Other: HIV exposure or maternal immunodeficiency.

**Figure 2 IJNS-09-00017-f002:**
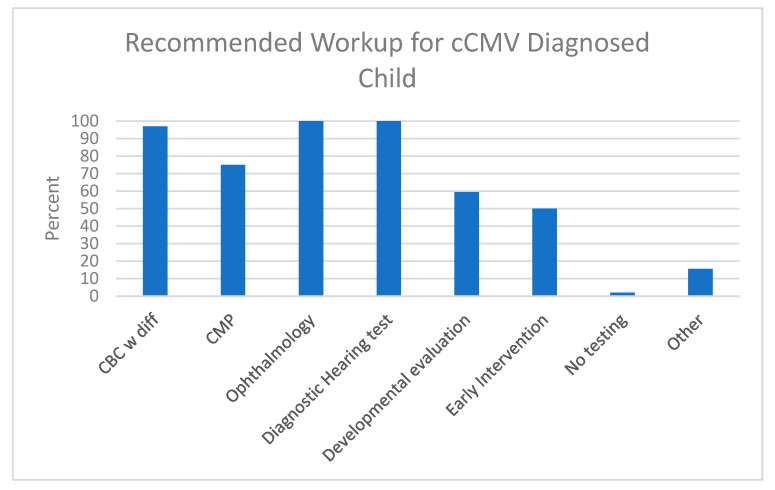
Recommended Workup for cCMV-diagnosed infant according to the PID respondents. Comprehensive metabolic profile (CMP): glucose, calcium, sodium, potassium, carbon dioxide, chloride, blood urea nitrogen, creatinine, albumin, protein, transaminases; other: abdominal ultrasound, CMV PCR blood titers and toxoplasmosis serology.

**Figure 3 IJNS-09-00017-f003:**
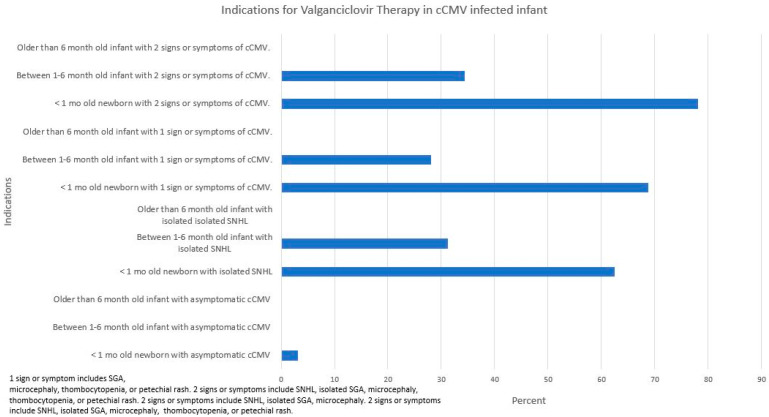
Indications for valganciclovir therapy in cCMV-infected infants.

**Table 1 IJNS-09-00017-t001:** Demographics of the PID Respondents.

Characteristic	Pediatric Infectious Disease (n = 32)
Age	49.0 years
	SD = 10.2
	range: 30–66
**Gender**	
Female	15 (46.9%)
Male	17 (53.1%)
**Race**	
Asian	7 (21.9%)
White	26 (81.3%) ^1^
**Years in Practice**	
0–5	4 (12.5%)
6–10	3 (9.4%)
11–15	6 (18.8%)
>15	16 (50%)
Still in training	3 (9.4%)
**Current Position**	
PID private practice	1 (3.1%)
PID academic	28 (90.3%)
PID fellow in training	3 (9.7%)
**% Children covered by Medicaid insurance in practice ^2^**	
0–25	2 (7.1%)
26–50	9 (32.1%)
51–75	6 (19.4%)
76–100	2 (7.1%)
Unsure	9 (32.1%)

^1^ One respondent claimed to be from 2 races. ^2^ Four respondents are from Canada which does not have a Medicaid program. PID: Pediatric Infectious Disease.

**Table 2 IJNS-09-00017-t002:** Knowledge of cCMV symptoms (%).

	Correct Responses		
Symptom	Selected	Not Selected	Percentage Correct
**Correct symptoms**			
Hearing Loss	32	0	100.0%
Intellectual disability	30	2	93.8%
Loss of Vision	30	2	93.8%
Seizures	28	4	87.5%
Petechiae	32	0	100.0%
**Incorrect symptoms**			
Oral cavity ulcers	0	32	100.0%

**Table 3 IJNS-09-00017-t003:** Knowledge of cCMV and hearing (%).

	Correct Responses		
Incidence	Selected	Not Selected	Percentage Correct
**Approximately 10% of children with asymptomatic cCMV will develop SNHL.**	27	5	84.4%
**Approximately 33% of children with** **symptomatic cCMV will develop SNHL**	12	20	37.5%
Approximately 30% of children with asymptomatic cCMV will develop SNHL.	4	28	87.5%
Approximately 95% children with symptomatic cCMV will develop SNHL.	3	29	90.6%

Note: Correct responses are in bold. SNHL= sensorineural hearing loss.

**Table 4 IJNS-09-00017-t004:** Knowledge of cCMV hearing loss presentation (%).

	Correct Responses	
% Progressive Hearing Loss	Selected	Percentage Correct
5	3	9.4
**20**	5	15.6
35	7	21.9
50	17	53.1

Note: Correct responses are in bold.

**Table 5 IJNS-09-00017-t005:** Knowledge of cCMV testing for diagnosis.

	Correct Responses		
	Selected	Not Selected	Percentage Correct
**Dried blood spot CMV PCR after 3 weeks of age.**	4	28	12.5%
Dried blood spot CMV PCR prior to 3 weeks of age.	22	10	31.2%
Urine PCR/culture at any age	0	32	100.0%
**Urine PCR/culture prior to 3 weeks of age.**	31	1	96.9%
Saliva PCR/culture at any age	1	31	96.9%
Saliva PCR/culture prior to 3 weeks of age	25	7	21.9%
Saliva CMV culture with confirmatory urine PCR or culture at any age.	0	32	100.0%
**Saliva CMV culture with confirmatory urine PCR** **or culture prior to 3 weeks of age.**	15	17	46.9%
Serologic CMV IgG testing at any age.	0	32	100.0%
Serologic CMV IgG testing prior to 3 weeks of age.	0	32	100.0%

Note: Correct responses are in bold.

**Table 6 IJNS-09-00017-t006:** Knowledge of timing for CMV diagnosis.

	Correct Responses		
Which Test(s) Can Definitively Establish a Diagnosis for cCMV in Children Greater than 3 Weeks of Age?	Selected	Not Selected	Percentage Correct
**Dried blood spot CMV PCR testing**	20	12	68.8%
Serology for CMV IgG	0	32	100.0%
Serology for CMV IgM	2	30	93.8%
Imaging studies including CT and MRI	7	25	78.1%
Urine PCR/culture for CMV	0	32	100.0%
Saliva culture for CMV	0	32	100.0%
Other (eye findings; history of maternal seroconversion and symptomatic infant)	2	30	93.8%

Note: Correct responses are in bold.

**Table 7 IJNS-09-00017-t007:** CMV transmission.

	Correct Responses		
Transmission Route	Selected	Not Selected	Percentage Correct
**Kissing**	31	1	96.9%
**Changing diapers without hand washing afterwards**	31	1	96.9%
**Drinking breast milk**	31	1	96.9%
**Receiving a blood transfusion**	31	1	96.9%
**Sexual intercourse**	25	7	78.1%
**Sharing food**	20	12	62.5%

Note: Correct responses are in bold.

**Table 8 IJNS-09-00017-t008:** Type of neuroimaging ordered.

	Responses		
Neuroimaging	Selected	Not Selected	Percentage
Head ultrasound (HUS) for all cases	11	21	34.4%
MRI brain for all cases	7	25	21.9%
HUS for all cases followed by brain MRI if an abnormality is found	22	10	68.8%
HUS except a brain MRI if infant has microcephaly	7	25	21.9%
Other (positive SNHL and/or abnormal neurologic finding	1	31	3.1%

**Table 9 IJNS-09-00017-t009:** Protocol for valganciclovir treatment.

Tests Ordered during Valganciclovir Treatment	Selected	Percentage
CBC with differential	28	87.5%
Viral titers	4	12.5%
Drug concentration or pharmacokinetic studies	0	0.0%
CMP	18	56.3%
Drug resistance	1	3.1%
Other	7	21.9%
Duration of Valganciclovir Administration	Selected	Percentage
6 weeks	0	0.0%
6 months	29	90.6%
9 months	0	0.0%
12 months	1	3.1%
Until urine viral titers are undetectable	0	0.0%
Other	2	6.3%

## Data Availability

Data are not available due to privacy considerations.

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
