# Peer review of "A Cross-Sectional Survey of Pediatric Infectious Disease Physicians’ Approach to Congenital Cytomegalovirus Infectionâ€"

_2409-515X, 2023, doi:10.3390/ijns9020017_

Round 1
Reviewer 1 Report
Review 1 of Hoki C ea.
The authors report the results of a survey using a questionnaire directed to Pediatric Infectious Disease (PID) physicians in North America concerning their knowledges and management of cCMV infection disease and policy of their health care institutions. The survey gives an interesting insight into countries, where congenital CMV infection is a well recognized entity by the health authorities with numerous scientific activities and relevant sponsoring. In addition, relevant questions concerning cCMV infection up to now unresolved by clinical trials are named.
There are only minor concerns:
1. I did not find the questionnaire, it would be helpful, if the questionnaire could be added in the e-supplement.
2. The authors should state, how many PID physicians have been contacted and how many institutions, and both from which country. This important information should be located at the beginning of the Method section. This could offer an idea to the reader about the resonance of the survey.
Line 4: omit “and”
Line 67ff: I would like to suggest to shorten: … PID physicians involved in the NIH funded ValEAR clinical trial (ClinicalTrials.gov NCT03107871). The study included … and please add here the number of PID physicians contacted.
L 74: more precise and complete, please, see above
L 91: 32 of how many PID phys. contacted?
L 148: inconsistencies in writing: number, comma, percentage
L 152: Fig. 1: Helpful, if the order of the indications to test for CMV would be improved: e.g. sorted by percentage of the specified item, with the same sorting in the Figure and in the text.
Actually it is like random.
The line "Other" should be located at the end .
The same is valid for the other Figures.
L 171: comprehensive metabolic profile: please add the included parameters.
L 204: Do you mean “asymptomatic cCMV infection”?
L 246: Does Ref. 19 fit?
L 252: I could not find Ref. 20 in Pubmed, nor by Google search.
L 264: This is an important pleading for a general SOP!
L 270: Please use “with” instead of “w”.
L 276: Perhaps the authors could add here the important dutch and canadian contributions: HUS and MRI are complementary (2022 03 Keymeulen A Cranial US and MRI complementary or not in the diagnostic assessment of children with cCMV Eur J Ped 2022; Vande Walle Brain MRI Findings In Newborns With cCMV Europ Radiol 2021; Smiljkovic M. Head ultrasound, CT or MRI 2019: The choice of neuroimaging in the assessment of infants with cCMV infection, BMC Pediatr. 2019.
L 352: Perhaps the authors could, in the conclusion section, add, that a general SOP is lacking and this could help avoiding different managements.
Reviewer 2 Report
I thank the authors for this interesting study. I work in the field of neonatal CMV and Europe and I think it is an infection that needs to be talked about as much as possible to raise awareness among the general population and the medical profession, especially on prevention issues.
Regarding the form, I invite authors to pay attention to the quality of their manuscript and to read it very carefully. There are differences in font size, duplication of words or % and some spaces are missing. A thorough proofreading must be done before publication. The figures are illegible because the resolution is too low and the tables could be greatly optimised and more aesthetically pleasing. The whole thing gives a bit of a sloppy impression and this is very detrimental to the overall quality of the article. I also invite the authors not to start their discussion by claiming a "first time" which should be reserved in my opinion for absolutely major scientific discoveries and is not at all appropriate for this type of study.
As far as the substance of the work is concerned. I do think it is a good idea to ask paediatricians about their knowledge and practice on congenital CMV. On the other hand, I don't quite understand the recruitment of participants: you talk about physicians involved in the ValEAR clinical trial, which implies that they are practitioners who are already aware. Why didn't you propose this on a wider scale? This should be discussed in the limitations of the study. The discussion is very long, perhaps it would be interesting to reduce it and highlight the most serious points revealed by this questionnaire and, above all, what possible improvements could be proposed?
Author Response
Reviewer #2:
Regarding the form, I invite authors to pay attention to the quality of their manuscript and to read it very carefully. There are differences in font size, duplication of words or % and some spaces are missing. A thorough proofreading must be done before publication. The figures are illegible because the resolution is too low and the tables could be greatly optimised and more aesthetically pleasing. The whole thing gives a bit of a sloppy impression and this is very detrimental to the overall quality of the article. I also invite the authors not to start their discussion by claiming a "first time" which should be reserved in my opinion for absolutely major scientific discoveries and is not at all appropriate for this type of study. Thank you for your comments. Apparently, the manuscript you reviewed was imported and altered from the manuscript we submitted. I believe the transfer process resulted in differences in font size, duplications of words and missing spaces. I have tried to correct those errors but would defer to the editors regarding the figure resolution and optimization of the tables.
As far as the substance of the work is concerned. I do think it is a good idea to ask paediatricians about their knowledge and practice on congenital CMV. On the other hand, I don't quite understand the recruitment of participants: you talk about physicians involved in the ValEAR clinical trial, which implies that they are practitioners who are already aware. Why didn't you propose this on a wider scale? Multiple attempts were made to try to reach more PID physicians. Unlike other organizations such as the American Academy of Otolaryngology, the American Society of Pediatric Otolaryngology or the American Speech-Language-Hearing association in which it is relatively easy to contact providers, there is not an infrastructure within the PID society for large scale surveys. That may explain why no prior survey of PID physicians on cCMV evaluation and management has been done before. Additional language was added to the limitations section of the discussion. This should be discussed in the limitations of the study. The discussion is very long, perhaps it would be interesting to reduce it and highlight the most serious points revealed by this questionnaire and, above all, what possible improvements could be proposed? We acknowledge that the discussion is long. For that reason, we broke up the sections to highlight different aspects of the survey responses and aid the reader. We believe that the elements in the discussion are important and highlight the key findings from the survey. We also feel that the International Journal of Neonatal Screening given its open access format that is not limited by journal space is the perfect media for this study.